# Prediction of Geometric Dimensions of Deposited Layer Produced Using Laser-Arc Hybrid Additive Manufacturing

**DOI:** 10.3390/mi15070830

**Published:** 2024-06-27

**Authors:** Junfei Xu, Junhua Wang, Yanming Wu, Xiaojun Liu, Jianjun Peng, Kun Li, Kui He, Tancheng Xie

**Affiliations:** 1School of Mechanical and Electrical Engineering, Henan University of Science and Technology, Luoyang 471003, China; mecha_xjf@163.com (J.X.); pjjsdu@163.com (J.P.); hekui@haust.edu.cn (K.H.); 2Henan Intelligent Manufacturing Equipment Engineering Technology Research Center, Luoyang 471003, China; 3Henan Engineering Laboratory of Intelligent Numerical Control Equipment, Luoyang 471003, China; 4Luoyang Ship Material Research Institute, Luoyang 471023, China; yanming_wu@163.com; 5Shenyang Aircraft Corporation, Shenyang 110000, China; liym@sg43.com; 6College of Mechanical and Vehicle Engineering, Chongqing University, Chongqing 400044, China; kun.li@cqu.edu.cn

**Keywords:** deposited layers, geometric dimensions, particle swarm optimization, extreme gradient boosting

## Abstract

Laser-arc hybrid additive manufacturing (LAHAM) holds substantial potential in industrial applications, yet ensuring dimensional accuracy remains a major challenge. Accurate prediction and effective control of the geometrical dimensions of the deposited layers are crucial for achieving this accuracy. The width and height of the deposited layers, key indicators of geometric dimensions, directly affect the forming precision. This study conducted experiments and in-depth analysis to investigate the influence of various process parameters on these dimensions and proposed a predictive model for accurate forecasting. It was found that the width of the deposited layers was positively correlated with laser power and arc current and negatively correlated with scanning speed, while the height was negatively correlated with laser power and scanning speed and positively with arc current. Quantitative analysis using the Taguchi method revealed that the arc current had the most significant impact on the dimensions of the deposited layers, followed by scanning speed, with laser power having the least effect. A predictive model based on extreme gradient boosting (XGBoost) was developed and optimized using particle swarm optimization (PSO) for tuning the number of leaf nodes, learning rate, and regularization coefficients, resulting in the PSO-XGBoost model. Compared to models enhanced with PSO-optimized support vector regression (SVR) and XGBoost, the PSO-XGBoost model exhibited higher accuracy, the smallest relative error, and performed better in terms of Mean Relative Error (MRE), Mean Square Error (MSE), and Coefficient of Determination R^2^ metrics. The high predictive accuracy and minimal error variability of the PSO-XGBoost model demonstrate its effectiveness in capturing the complex nonlinear relationships between process parameters and layer dimensions. This study provides valuable insights for controlling the geometric dimensions of the deposited layers in LAHAM.

## 1. Introduction

As the manufacturing industry advances, there is an escalating demand for manufacturing processes that offer enhanced efficiency and stability, particularly for the production of complex components. Traditional laser and arc additive manufacturing technologies are facing increasing challenges in meeting these requirements [1]. Laser-arc hybrid additive manufacturing (LAHAM) technology addresses this gap by combining the strengths of both laser and arc additive manufacturing, offering higher molding efficiency, more stable processing outcomes, robust adaptability, commendable mechanical properties, and high raw material utilization [2]. Its extensive applications span across aerospace, rail transportation, automotive manufacturing, and other industrial sectors [3]. The geometric dimensions of LAHAM-deposited layers are pivotal for assessing layer quality, as they are closely related to the precision of the formative processes and the mechanical properties of the deposited material. Achieving precise control over these dimensions is fundamental for fabricating high-quality and high-performance components [4]. However, the LAHAM process is a complex multi-field coupling phenomenon, with the geometric dimensions of the deposited layers being influenced by a multitude of process parameters. The current lack of clarity regarding the influence and mapping relationships of these parameters on the deposited layer’s geometric dimensions presents a significant challenge for precise control. The uncertainty in the geometric dimensions of deposited layers is a major impediment to the widespread adoption of LAHAM. Accurate prediction of these dimensions is essential for minimizing the need for extensive post-processing, thereby reducing production cycles and enhancing manufacturing efficiency. Moreover, meticulous prediction and control of deposition layers are crucial for ensuring that the final product adheres strictly to design specifications, significantly improving the accuracy of the forming process. Inaccuracies in deposition can lead to defects, such as gaps or overlaps between layers, affecting the structural integrity of the manufactured component. The ability to predict layer deposition with high accuracy is vital for preventing such anomalies. Investigating the impact of various process parameters on the geometric dimensions of the deposited layer and establishing a mapping relationship between these parameters and the layer dimensions is crucial for achieving precise predictions of the deposited layer sizes. This endeavor is essential for enhancing the molding precision of LAHAM technology and improving manufacturing efficiency.

Laser-arc hybrid additive manufacturing has garnered extensive attention from both domestic and international researchers owing to its distinctive advantages, resulting in a multitude of research endeavors dedicated to this process. Wang et al. [5] conducted a comprehensive study on the control of molten pool geometry in the plasma transfer arc (PTA)-laser hybrid welding process. This study systematically revealed the impact patterns of various process parameters such as laser power, arc-laser separation distance, laser beam size, and arc current on the molten pool. Wojciech et al. [6] devised and subsequently verified a three-dimensional steady-state finite element model featuring dual independent circular surface heat sources. Experimental results unequivocally affirmed the model’s precision in predicting essential aspects such as melt pool geometry, heat-affected zones, and thermal cycles. Kapil et al. [7] developed an innovative deposition layer profile model for laser arc hybrid additive manufacturing and validated its accuracy through experimental procedures. The experimental results confirmed that the model had higher precision compared to conventional models. It was also found that scanning speed was the most critical factor affecting the deposition layer profile, followed by welding current. Liu et al. [8] prepared aluminum alloy thin-walled parts by hybrid additive manufacturing using electric arc and laser arc and analyzed their microstructure, phase structure, microhardness, and tensile properties. The microhardness, tensile strength, and elongation of the hybrid additive manufacturing were found to be improved to some extent by data analysis. Gong et al. [9] investigated the effect of laser power on the formation, organization evolution, and mechanical properties of 316 L stainless steel fabricated by laser-arc hybrid additive manufacturing. The results show that with the increase in laser power, the stability of the molten pool first rises and then decreases, and the surface accuracy first increases and then decreases. Kochar et al. [10] developed an optimized neural network topology. They used welding speed, wire feed speed, arc current, and laser power as inputs, with the polar coordinates of the weld bead profile as outputs. The results demonstrated the accurate prediction of the weld’s geometric contour. Ye et al. [11] utilized a high-speed camera to monitor the laser–metal inert gas arc welding (MIG) hybrid welding process. They fed the visual information into their model and designated the weld bead width as the model’s output. An attention-based long short-term memory (ATT-LSTM) predictive model was developed to forecast the width of the weld bead. The experimental outcomes demonstrated that this model possessed good predictive accuracy and generalization capabilities. Lei et al. [12] extracted various morphological features from the images of the melt pool and used principal component analysis (PCA) to diminish the redundancy of these features. The inputs for the model included welding speed, laser power, and two principal components from PCA. The outputs were the height and width of the deposited layer, with the neural network being optimized using a genetic algorithm (GA). The experimental results indicated that the model was capable of accurately predicting the geometric characteristics of the weld bead. In summary, current research on the impact of process parameters on the formation patterns of the deposited layer is primarily focused on the field of laser-arc hybrid welding [13,14,15]. Both laser-arc hybrid welding and laser-arc hybrid additive manufacturing share certain characteristics, such as the inverse relationship between the width and depth of the melt pool and the travel speed [16]. However, due to the different requirements of laser-arc hybrid welding and laser-arc hybrid additive manufacturing, there are significant differences in the development of formation rules and the selection of process parameters for these two distinct processes. To achieve precise control over the geometric dimensions of the deposited layer, it is essential to conduct an in-depth investigation into the effects of process parameters in laser-arc hybrid additive manufacturing on the said geometry and to establish an accurate predictive model for predicting the morphology and dimensions of the deposited layer.

In order to uncover the influence patterns of various process parameters on the deposited layer and to accurately predict the morphology and dimensions of the deposited layer, this study conducted process experiments to explore how these parameters affected layer dimensions and performed a quantitative analysis to evaluate the extent of their impact. By employing the particle swarm optimization (PSO) algorithm to optimize the hyperparameters of the XGBoost model, a PSO-XGBoost predictive model was developed, which is capable of precisely forecasting the geometric dimensions of the deposited layers. This research provides decision-making references and guidance for improving the quality and efficiency of the forming process.

## 2. Materials and Methods

### 2.1. Materials and Setup

In this study, experiments were conducted using a laser-arc hybrid additive manufacturing system, which mainly consists of a laser device, an arc apparatus, and a cooling system. The arc power source utilized was the Kaierda D350S (Hangzhou Kaierda Welding Robot Co., Ltd., Hangzhou, China), and the laser was provided by an RFL-2000 W laser (Wuhan Raycus Fiber Laser Technologies Co., Ltd., Wuhan, China). Both the welding head and the laser head were mounted on a Yaskawa GP25 robotic arm (YASKAWA Electric Corporation, Kitakyushu Fukuoka, Japan). A mixture of argon and carbon dioxide gases was used as the shielding gas at a flow rate of 6 L/min (Henan University of Science and Technology, Luoyang, China). The welding head was positioned at a 45° angle to the scanning direction, while the laser head was perpendicular to the substrate, ensuring perpendicular laser incidence. A water cooler was connected to the laser head to prevent equipment from overheating and potential damage during processing (Sanhe Tongfei Refrigeration Co., Ltd., Langfang, China). The schematic diagram of the aforementioned system is shown in Figure 1.

In the conducted experiment, the Keyence LG-8000A line laser scanner (Keyence Corporation, Osaka, Japan) was employed to acquire real-time measurements of both the width and height of the deposited layer. The outcomes obtained from the line laser scanner are visually presented in Figure 2.

In this study, the chosen substrate material was 45 steel. To mitigate potential sources of interference, such as heat accumulation arising from repetitive processing, a singular experiment was executed for each substrate, thereby ensuring the integrity of the experimental outcomes. Prior to conducting the experiment, the surface of the substrate underwent a preparatory procedure involving meticulous sandpaper polishing, which was succeeded by a comprehensive cleansing process aimed at eliminating any impurities by employing anhydrous ethanol. In this study, THQ-50C welding wire (Tianjin Daqiao Welding Materials Group Co., Ltd., Tianjin, China), a high-quality deposition material manufactured using advanced drawing and copper plating techniques, was employed. THQ-50C welding wire is a type of 500MPa-grade carbon steel wire known for its numerous advantages, including minimal spattering, high deposition efficiency, excellent deposition layer formation, and low susceptibility to metal porosity. It finds widespread application in various industries, such as engineering machinery, shipbuilding, and petrochemicals. In this experiment, a welding wire with a diameter of 0.8 mm was utilized. The chemical composition of the THQ-50C welding wire is detailed in Table 1.

### 2.2. Design of Experiments

To investigate the impact of various process parameters on the geometric dimensions of deposition layers, this research utilizes a controlled variable approach for process experimentation. Laser power, arc current, and scanning speed are selected as the independent variables, while the width and height of the deposition layer are considered dependent variables. This study entails a sequential alteration of the independent variables from low to high, maintaining other parameters constant. Preliminary experiments were conducted, and the analysis of the results revealed that when the laser power and arc current were low and the scanning speed was high, the input heat was insufficient to melt the material, resulting in poor forming quality. Conversely, when the laser power and arc current were high, and the scanning speed was low, excessive input heat leds to defects such as deformation and poor stability of the molten pool, resulting in suboptimal forming quality. Based on the preliminary experiment results, a reasonable range of process parameters was selected, as shown in Table 2.

## 3. Exploration of Deposition Layer Forming Law

Analysis of monitoring data procured from the line laser scanner revealed anomalous measurements in the height and width at the initial and terminal segments of the deposition layers. This irregularity can be attributed to the necessity of initiating and stabilizing the arc at the commencement of the process, which demands a significant initial energy input. This initiation phase results in an escalated heat input, leading to an expansion of the molten pool and, consequently, an enlargement in the dimensions of the deposited layer. Additionally, a sudden change in arc current and heat input at the termination of the deposition process induces a collapse of the layer. To mitigate the negative impact of these outliers on the experimental results, the data processing procedure involved selecting only the mean values of the width and height from the stabilized area for further analysis.

### 3.1. Exploration of Deposition Layer Width Forming Law

An analysis was conducted on the effects of various process parameters on the deposition layer. The variations in the width of the deposition layer with changes in process parameters are illustrated in Figure 3.

As shown in Figure 3, the effects of different process parameters on the width of the deposition layer are not consistent. When the arc current and scanning speed are held constant, the width of the deposition layer increases with an increase in laser power. Conversely, when the laser power and arc current remain constant, the width of the deposition layer decreases as the scanning speed increases. Keeping the laser power and scanning speed constant, the width of the deposition layer increases with an increase in arc current.

An analysis of the causes of this phenomenon reveals that when laser power increases, the energy input per unit time into the melt pool is higher, leading to a rise in melt pool temperature and, consequently, an expansion of the melt pool area. Simultaneously, this results in higher local temperatures and larger temperature gradients. According to the Marangoni Effect [17], as the temperature gradient of the melt pool increases, the surface tension gradient also increases. The molten metal flows from areas of lower surface tension (hotter regions) to areas of higher surface tension (cooler regions), causing the melt pool to widen. An increase in arc current increases the amount of material melted per unit time, which is a primary factor in the widening of the deposition layer. Additionally, a higher current increases the heat input into the melt pool per unit time, raising the temperature and size of the melt pool. When the laser power remains constant and the scanning speed increases, the heat input into the melt pool per unit time decreases, resulting in a smaller melt pool. Moreover, an increase in scanning speed accelerates cooling, leading to quicker solidification of the melt pool and limiting its flow. As the feed rate of the material remains constant, an increase in scanning speed means less material is deposited per unit length, directly leading to a reduction in the width of the deposition layer.

Increasing the laser power leads to an increase in the depth and diameter of the laser keyhole, which, in turn, increases the size of the molten pool and affects the width of the deposition layer. Reducing the scanning speed increases the dwell time of the laser on the material surface, resulting in a deeper and wider keyhole, which also enlarges the molten pool size. When the arc current increases, the total input heat of the system rises, causing the molten pool temperature to increase and thereby affecting the keyhole depth and shape. Higher input heat helps to form a deeper, more stable keyhole and enlarges the molten pool size, consequently increasing the dimensions of the deposition layer.

To investigate the extent to which different process parameters affect the width of the deposition layer, process parameters were used as independent variables and deposition layer width as the dependent variable. A nonlinear least squares method—the Marquardt method—was employed to perform multivariate nonlinear fitting on experimental data. The resulting objective function for the deposition layer width with respect to laser-arc hybrid additive manufacturing process parameters is as follows:(1)$$W=2.447\left(P^{0.027}+I^{0.355}+V^{-0.326}\right)$$

From the objective function, it is evident that the arc current has the most significant impact on the width of the deposition layer, followed by the scanning speed. The influence of laser power on the deposition layer width is the least.

As shown in Figure 3b, when the laser power is reduced, its impact on the width of the deposition layer decreases. To explain this phenomenon, the concept of laser energy density [18], *E*, is introduced. The formula for calculating laser energy density is as follows:(2)$$E=\frac{P}{D\cdot V}$$
where *P* represents the laser power; *V* represents the scanning speed, and *D* represents the area of the laser beam spot.

The trend of the deposition layer width as a function of laser energy density is shown in Figure 4d. At lower levels of laser energy density, the changes in energy density are insufficient to significantly alter the temperature of the melt pool. Consequently, the variations in the surface tension of the melt pool are limited, resulting in minimal changes in the geometric dimensions of the deposition layer.

### 3.2. Exploration of Deposition Layer Height Forming Law

An analysis was conducted on the effects of various process parameters on the deposition layer. The variations in the width of the deposition layer with changes in process parameters are illustrated in Figure 4.

As shown in Figure 4, when the arc current and scanning speed are held constant, the height of the deposition layer decreases with an increase in laser power. When the laser power and arc current remain unchanged, the width of the deposition layer decreases as the scanning speed increases. Keeping the laser power and scanning speed constant, the width of the deposition layer increases with an increase in arc current.

To explore the extent to which different process parameters influence the width of the deposition layer, process parameters were used as independent variables and deposition layer width as the dependent variable. A nonlinear least squares method—specifically, the Marquardt method—was employed to perform multivariate nonlinear fitting of the experimental data. The resulting objective function relating deposition layer width to laser-arc hybrid additive manufacturing process parameters is as follows:(3)$$H=1.253(P^{-0.011}+I^{0.732}+V^{-0.698})$$

From the objective function, it is apparent that the arc current has the greatest impact on the width of the deposition layer, followed by the scanning speed. The influence of laser power on the deposition layer width is the least.

## 4. Predictive Model for the Geometric Dimensions of Deposited Layers

### 4.1. Construction of a PSO-XGBoost Predictive Model

From the analysis in Section 3, it is evident that the relationship between the geometric dimensions of deposited layers in laser-arc hybrid additive manufacturing and various parameters, such as laser power, arc current, and scanning speed, is nonlinear and complex. The use of empirical formulas to establish mathematical, analytical expressions or simple linear regression modeling fails to accurately map the relationship between process parameters and the geometric dimensions of deposited layers. The XGBoost model, with its strong capability to handle nonlinear data, can effectively establish this mapping for accurate predictions [19]. The built-in regularization feature of XGBoost aids in preventing model overfitting. This means that it can provide accurate predictions while maintaining model complexity, which is particularly important for predicting the geometric dimensions of deposited layers. Unlike other machine learning algorithms, XGBoost offers a degree of interpretability, allowing for a better understanding of the key factors influencing the geometric dimensions of deposited layers. Moreover, given the inevitability of anomalies in the laser-arc additive manufacturing process, the XGBoost algorithm exhibits high robustness to outliers. Its tree-based models are generally not overly influenced by outliers, as the tree structure does not excessively depend on extreme values during node splitting. This robustness makes XGBoost particularly well-suited for predicting the geometric dimensions of deposited layers in laser-arc hybrid additive manufacturing. However, the prediction outcomes of the XGBoost algorithm are significantly influenced by its hyperparameters, including the learning rate and regularisation coefficient. The conventional approach of manually configuring these hyperparameters heavily depends on human expertise, which may not guarantee optimal accuracy and stability [20]. The PSO algorithm is a global stochastic search algorithm known for its simple structure and excellent global search capabilities [21]. It may be effectively applied to optimize hyperparameters in XGBoost.

Employing XGBoost for geometric dimensions of deposited layers prediction, the dataset comprises the process parameters and the corresponding geometric dimensions, which may be expressed as [22]:(4)$$A=\left\{X_{mn},Y_{m}\right\}$$
where $X_{mn}$ represents the process parameter; $Y_{m}$ represents the geometric dimensions of the deposited layer; *m* represents the number of samples, and *n* represents the number of process parameters. $X_{mn}$ is an input to XGBoost, and $Y_{m}$ is an output from XGBoost. After establishing the sample dataset, the predictive model can be expressed as follows:(5)$$F(x_{i})=f_{0}(x_{i})+\overset{K_{X}}{\underset{K=1}{\Sigma }}\overset{T_{k}}{\underset{j=1}{\Sigma }}\omega_{j,k}\cdot \eta$$
where $f_{0}(x_{i})$ is the initial model; $x_{i}$ is the ith sample; k=1,2,3,⋅⋅⋅ is the number of iterations; j=1,2,3,⋅⋅⋅, $T_{k}$ is the number of CART leaf nodes and the number of CART leaf nodes in the *k*th iteration; $\omega_{j,k}$ is the alternative value of all samples corresponding to the jth leaf node in the kth iteration; η is the learning rate.

$f_{0}(x_{i})$ is generally determined by the loss function as shown in Equation (6):(6)$$f_{0}(x_{i})=\underset{\cdot }{\arg\min}\sum_{i=1}^{m}L(y_{i},\alpha )$$
where $L(y_{i},\alpha )$ represents the loss function, which quantifies the deviation between the target values and the predicted values. $y_{i}$ represents the target value for the *i*-th sample, and α is the constant that minimizes the loss function. $\omega_{j,k}$ is determined by Equation (7), which is given below:(7)$$\omega_{j.k}=-\frac{\sum \partial L(y_{i},{\overset{\wedge }{y}}_{i,k-1})/\partial {\overset{\wedge }{y}}_{i,k-1}}{\sum \partial^{2}L({\overset{\wedge }{y}}_{i,k-1})/\partial {\overset{\wedge }{y^{2}}}_{i,k-1}+\lambda }$$
where ${\overset{\wedge }{y}}_{i,k-1}$ is the predicted value of the i sample at the *k*−1st iteration; λ is the regularization parameter.

Utilizing various loss functions in XGBoost yields distinct training outcomes and exerts a significant influence on the prediction results. The use of the squared loss function makes the model sensitive to outliers or noise points in the data. In this study, the laser-arc hybrid additive manufacturing process is influenced by various factors, and even minor environmental fluctuations can lead to variations in the deposited layer, resulting in the presence of outliers. Therefore, we employ the absolute loss function for modeling.

The PSO optimization algorithm is employed to optimize the number of leaf nodes, learning rate, and regularization parameter of the XGBoost model. Setting the population size to q, the XGBoost parameter population can be expressed as follows:(8)$$\Gamma =\left\{\sigma_{i^{’}}|\sigma_{i^{’}}=(\sigma_{1i^{’}},\sigma_{2i^{’}},\cdot \cdot \cdot ,\sigma_{zi^{’}})\;\;i^{’}=1,2,\cdot \cdot \cdot ,q\right\}$$
where $\sigma_{i^{’}}$ represents the seed vector containing *z* optimization parameters.

The PSO iteration formula can be expressed as follows [23]:(9)$$\Delta_{i^{’}}^{j^{’}}=\left\{\sigma_{i^{’}}^{j^{’}}|\sigma_{i^{\prime }}^{j^{\prime }}=\min({\left\|O_{i^{\prime }}^{j^{\prime }}-O_{std}\right\|}_{2})\;\;j^{\prime }=1,2,\cdot \cdot \cdot ,t\right\}\;\Delta_{g}^{j^{\prime }}=\left\{\sigma_{\Gamma }^{j^{\prime }}|\sigma_{\Gamma }^{j^{\prime }}=\min({\left\|O_{\Gamma }^{j^{\prime }}-O_{std}\right\|}_{2})\right\}\;W_{i^{\prime }}^{j^{\prime }+1}=\omega W_{i^{\prime }}^{j^{\prime }}+c_{1}r_{1}(\Delta_{i^{\prime }}^{j^{\prime }}-\sigma_{i^{\prime }}^{j^{\prime }})+c_{2}r_{2}(\Delta_{g^{\prime }}^{j^{\prime }}-\sigma_{i^{\prime }}^{j^{\prime }})\;\sigma_{i^{\prime }}^{j^{\prime }+1}=\sigma_{i^{\prime }}^{j^{\prime }}+W_{i^{\prime }}^{j^{\prime }+1}\;s.t.\;\begin{array}{c} \sigma_{i^{\prime }}^{j^{\prime }}\in \left[\sigma_{\max\_1},\sigma_{\max\_2}\right]\\ W_{i^{\prime }}^{j^{\prime }}\in \left[-W_{\max},W_{\max}\right] \end{array}$$
where $\Delta_{i^{\prime }}^{j^{\prime }}$ is the minimum computational error of the $i^{\prime }$th seed at the $j^{\prime }$th iteration; $\Delta_{g^{\prime }}^{j^{\prime }}$ is the seed that has the lowest error among the previous $j^{\prime }$ iterations; $\sigma_{i^{\prime }}^{j^{\prime }}$represents the $i^{\prime }$-th seed at the $j^{\prime }$-th iteration; $\sigma_{\Gamma }^{j^{\prime }}$ is any one of the seeds at iteration $j^{\prime }$; $O_{i^{\prime }}^{j^{\prime }}$represents the computation result of the $j^{\prime }$-th iteration for the $i^{\prime }$-th seed; $O_{std}$ is the result of the target calculation; $O_{\Gamma }^{j^{\prime }}$ is the result of the computation of any seed at the $j^{\prime }$-th iteration; $O_{\Gamma }^{j^{\prime }}$ is the result of the computation of any seed at the $j^{\prime }$-th iteration; t is the number of iterations; $W_{i^{\prime }}^{j^{\prime }}$ is the step size of the $i^{\prime }$-th seed at the $j^{\prime }$-th iteration; ω is the gravity coefficient; $c_{1},c_{2}$ is the acceleration factor; $r_{1},r_{2}$ is the elasticity coefficient; $\sigma_{\max\_1},\sigma_{\max\_2}$ is the seed parameter value bounds; $W_{\max}$ is the maximum bound on the iteration step size.

To lock in the direction of the global optimal solution in the early stages of PSO iteration, a generally large value ω is adopted, while a smaller value is used for local search in the later stages of iteration. This study utilizes a linear optimization method to optimize ω.

The diagram in Figure 5 illustrates the sequence of steps involved in the PSO-XGBoost deposition layer geometric dimensions prediction model.

### 4.2. Analysis of Predictive Outcomes

This study includes a total of 125 experimental data sets, which are divided into training and testing datasets. The training dataset is used for model training, and the testing dataset is used to evaluate the model’s performance. A random selection of 110 data sets is used as the training dataset, while 15 data sets serve as the testing dataset. To further verify the accuracy and effectiveness of the PSO-XGBoost prediction model, a SVR model was also established as a comparative prediction model. The PSO algorithm was used to optimize the penalty factor C and the kernel parameter g of the SVR model, enabling it to achieve optimal predictive performance. The process of optimizing the SVR algorithm using the PSO algorithm is illustrated in Figure 6.

Predictions were made using the PSO-SVR model, the XGBoost model, and the PSO-XGBoost model, and the results were compared. To provide a clearer demonstration of the prediction accuracy of each model and to scientifically and objectively evaluate their performances, this paper utilizes the mean relative error (MRE), mean square error (MSE), and coefficient of determination (R^2^) as quantitative evaluation indices. $\Delta_{E}$ is defined as the estimated relative error between the predicted and actual values of each model.
(10)$$R^{2}=\frac{\sum_{i=1}^{n}{\left(\hat{y_{i}}-\overset{-}{y}\right)}^{2}}{\sum_{i=1}^{n}{\left(y_{i}-\overset{-}{y}\right)}^{2}}$$
(11)$$\mathrm{MSE}=\frac{1}{n}\sum_{i=1}^{n}{\left(\hat{y_{i}}-y_{i}\right)}^{2}$$
(12)$$\mathrm{MRE}=\frac{1}{n}\frac{\sum_{i=1}^{n}\left|\hat{y_{i}}-y_{i}\right|}{y_{i}}$$
(13)$$\Delta_{E}=\left|\frac{y_{i}-\overset{\wedge }{y_{i}}}{y_{i}}\right|\times 100\%$$
where ${\hat{y}}_{i}$ represents the *i*-th predicted value of the test set; $y_{i}$ denotes the *i*-th true value of the test set; $\bar{y}$ represents the average of the true values of the test data set.

#### 4.2.1. Prediction of Deposited Layers Width

The models were trained using a training set and subsequently tested using a test set. The comparisons between the predicted values and actual values for each model on the test set are shown in Figure 7.

Figure 7a reveals that the predictions from the PSO-XGBoost model are closest to the actual values, whereas the PSO-SVR and XGBoost models exhibit significant deviations from the actual values. Additionally, the predictions from the PSO-SVR and XGBoost models show greater variability, resulting in poorer prediction accuracy and stability. According to Figure 7b, the PSO-XGBoost model demonstrates the smallest relative error, with a maximum relative error of 5.39%. In contrast, the maximum relative errors for the PSO-SVR and Ridge Regression models are 14.23% and 14.5%, respectively.

The quantitative evaluation metrics for the predictive accuracy of each model are presented in Table 3.

In this study, the PSO-XGBoost model exhibited a coefficient of determination closest to 1, indicating its strong ability to fit the width of the deposited layers accurately. The MSE measures the magnitude of prediction errors in regression models; a smaller MSE indicates smaller prediction errors. As shown in Table 3, the MSE of the PSO-XGBoost model is significantly lower than that of the other two models. MRE measures the relative error in regression models, with a lower MRE indicating less relative error. The MRE of the PSO-XGBoost model is considerably smaller than those of the other two models.

#### 4.2.2. Prediction of Deposited Layers Height

Figure 8 displays a comparison between the projected and true values of the height of the deposited layers for each model.

Figure 8a shows that the predictions from the PSO-XGBoost model are closest to the actual values, while the predictions from the PSO-SVR and XGBoost models differ more significantly and exhibit poorer stability, failing to accurately predict changes in the height of the deposited layers. Figure 8b provides a more visual representation, where the relative errors between the predicted values and the actual values by the PSO-XGBoost model are the smallest, indicating the highest prediction accuracy.

The quantitative evaluation metrics for the predictive performance of each model are presented in Table 4.

Table 4 reveals that the coefficient of determination for the PSO-XGBoost model is closest to 1, indicating that this model accurately fits the height of the deposited layers. Additionally, the MSE and MRE for the PSO-XGBoost model are significantly lower than those for the other two models.

In summary, the PSO-XGBoost prediction model outperforms other models in all evaluated metrics, exhibiting the smallest relative error and highest predictive accuracy. It provides more accurate predictions of the geometric characteristics of the deposited layers, offering greater reference value.

## 5. Conclusions

To investigate the effects of various process parameters on the morphology and dimensions of the deposited layers in the laser-arc hybrid additive manufacturing process and to accurately predict the dimensions of the deposited layers, this study conducted extensive experiments and performed a quantitative analysis of the influence patterns of process parameters on the deposited layers. Furthermore, a particle swarm optimization–eXtreme gradient boosting (PSO-XGBoost) predictive model was established. Based on experimental data and model predictions, the following conclusions were drawn:
The width of the deposited layers increases with the rise of laser power and arc current and decreases as the scanning speed increases. The height of the deposited layers decreases with the increase in laser power and scanning speed and increases with the rise of arc current. Quantitative analysis using the Marquardt method reveals that the arc current has the most significant impact on the dimensions of the deposited layers, followed by the scanning speed, while the influence of laser power is the least.Based on the experimental results, the PSO-XGBoost model for predicting the dimensions of the deposited layers was constructed using laser power, scanning speed, and arc current as input variables and the width and height of the deposited layers as output variables. Additionally, the PSO-SVR model and the XGBoost model were used as comparative models. These models were trained using a training dataset and then tested for performance using a test dataset. A comparison of the predictive results from each model showed that the PSO-XGBoost model outperformed the others across all metrics, accurately forecasting the dimensions of the deposited layers. This approach provides effective guidance for enhancing the precision of the forming process.

## Figures and Tables

**Figure 1 micromachines-15-00830-f001:**
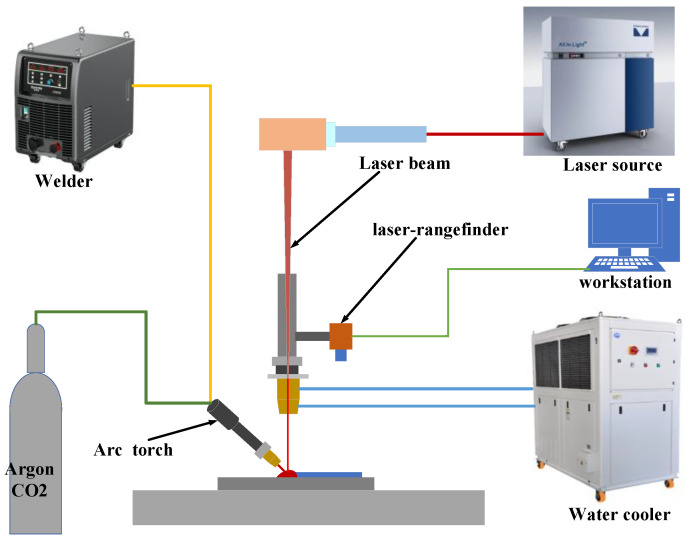
Schematic diagram of the laser-arc hybrid additive manufacturing system.

**Figure 2 micromachines-15-00830-f002:**
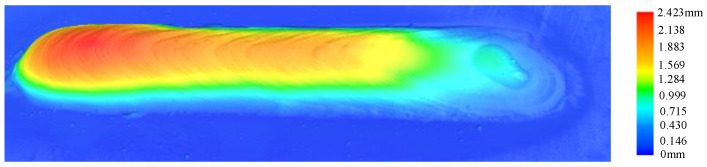
Results of 3D imaging Using a line laser scanner.

**Figure 3 micromachines-15-00830-f003:**
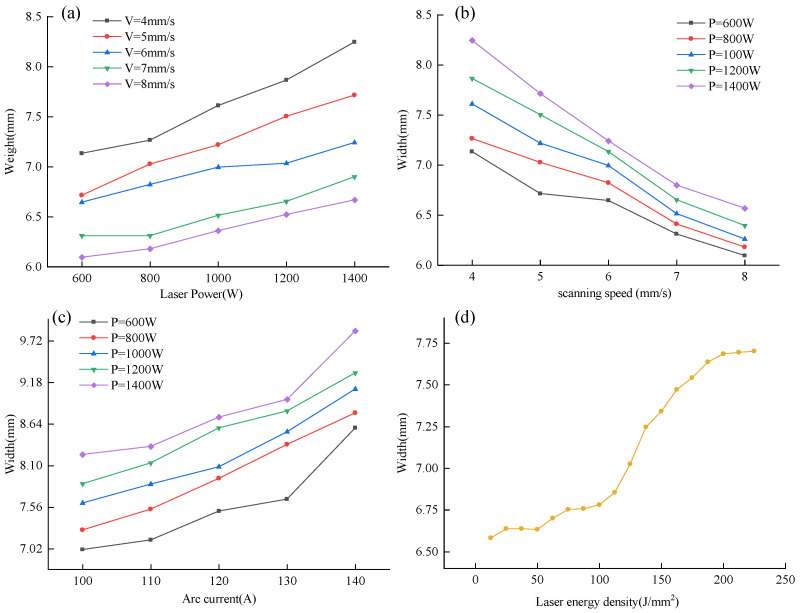
Variations in deposition layer width with process parameters. (**a**) The width of the deposited layer varies with the laser power; (**b**) The width of the deposited layer changes with the scanning speed; (**c**) The width of the deposition layer varies with the arc current; (**d**) The width of the deposited layer varies with the laser energy density.

**Figure 4 micromachines-15-00830-f004:**
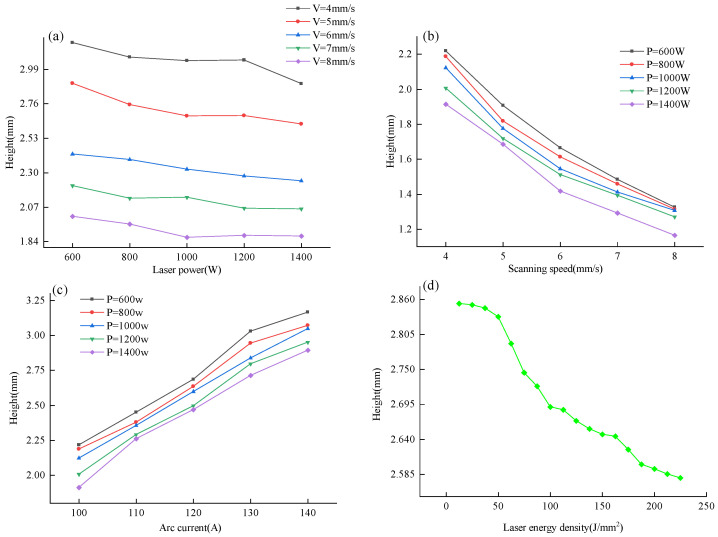
Variations in deposition layer height with process parameters. (**a**) The height of the deposited layer changes with the laser power; (**b**) The height of the deposition layer varies with the scanning speed; (**c**) The height of the deposition layer varies with the arc current; (**d**) The height of the deposited layer varies with the laser energy density.

**Figure 5 micromachines-15-00830-f005:**
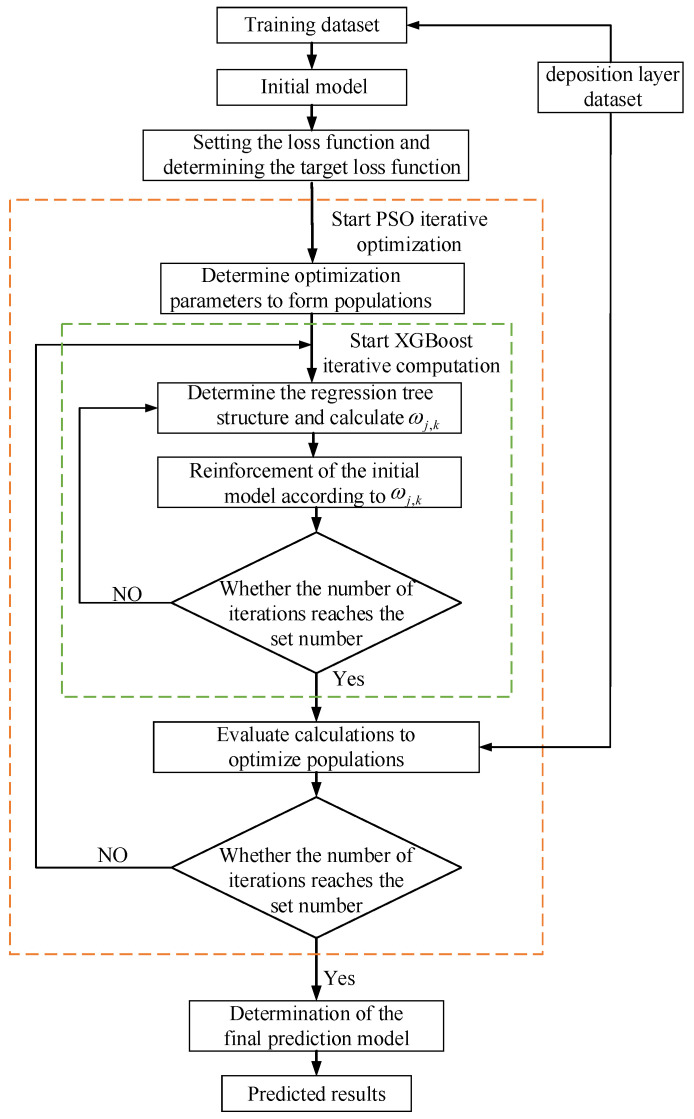
PSO optimisation XGBoost flowchart.

**Figure 6 micromachines-15-00830-f006:**
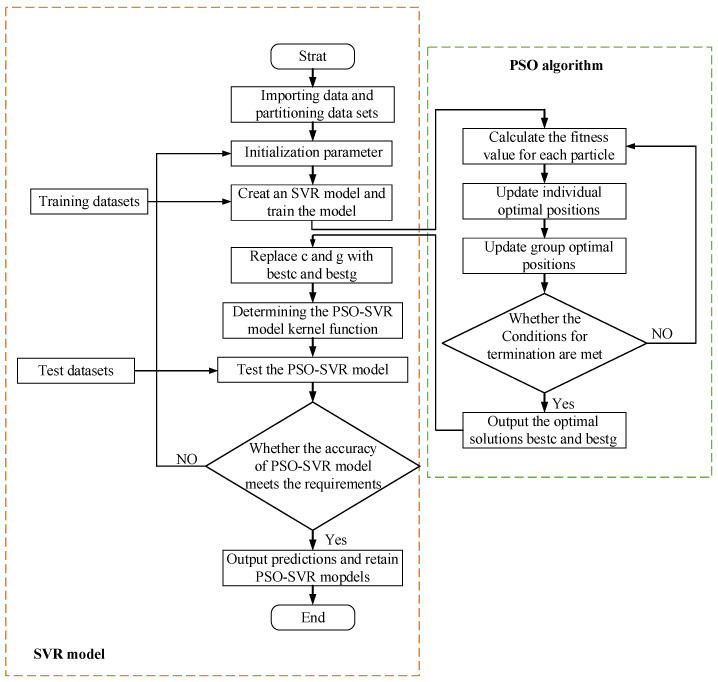
PSO optimization SVR flowchart.

**Figure 7 micromachines-15-00830-f007:**
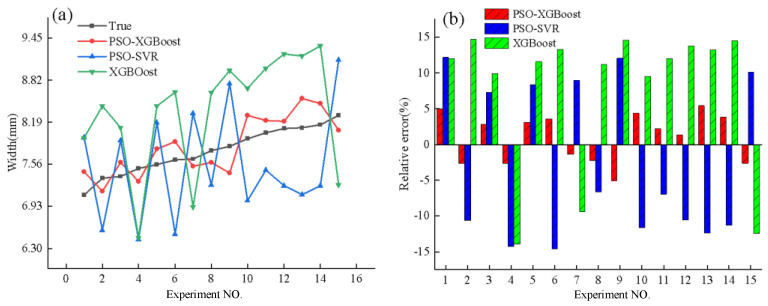
Comparison of predicted and true values of deposited layer width from different models. (**a**). Comparison of predicted and actual values across different models. (**b**). Relative error between predicted and actual values in various models.

**Figure 8 micromachines-15-00830-f008:**
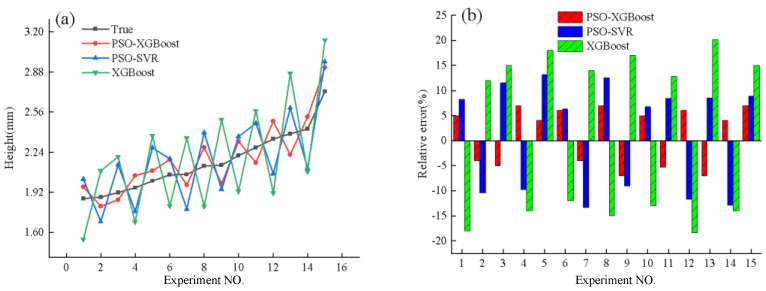
Comparison of predicted and true values of deposited layer height from different models. (**a**). Comparison of predicted and actual values across different models. (**b**). Relative error between predicted and actual values in various models.

**Table 1 micromachines-15-00830-t001:** Chemical composition of THQ-50C.

Element	C	Mn	Si	S	P	Cr	Ni	Cu	Nb	V
Wt%	0.08	1.50	0.89	0.012	0.013	0.02	0.03	0.11	0.02	0.003

**Table 2 micromachines-15-00830-t002:** Design of experiments.

Process Parameters (Symbol, Unit)	Value
laser power (*P*, W)	600, 800, 1000, 1200, 1400
scan speed (*v*, mm/s)	4, 5, 6, 7, 8
welding current (*I*, A)	100, 110, 120, 130, 140
Argon gas flux (*Q*, L/min)	8
laser spot diameter (*d*, mm)	2

**Table 3 micromachines-15-00830-t003:** Predictive Performance Metrics for Deposited Layer Width in Various Models.

	PSO-XGBoost	PSO-SVR	XGBoost
R^2^	0.959	0.824	0.671
MSE	0.004	0.092	0.148
MRE	0.006	0.031	0.076

**Table 4 micromachines-15-00830-t004:** Predictive performance metrics for deposited layer height in various models.

	PSO-XGBoost	SVR	Ridge Regression
R^2^	0.973	0.875	0.812
MSE	0.002	0.008	0.011
MRE	0.013	0.022	0.041

## Data Availability

Data are contained within the article.

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
