# Peer review of "Prediction of Geometric Dimensions of Deposited Layer Produced Using Laser-Arc Hybrid Additive Manufacturing"

_micromachines, 2024, doi:10.3390/mi15070830_

Round 1
Reviewer 1 Report (Previous Reviewer 2)
Comments and Suggestions for Authors
I still cannot see any scientific value in the presented manuscript.
The predictions that can be made by the suggested model is only valid for the exact parametrization of the experimental setup used. Generalization into other situations using Laser-arc hybrid additive manufacturing is not possible.
The manuscript still contains significant errors that have not been corrected.
Comments on the Quality of English LanguageLanguage is OK
Author Response
Please see the attachment.

Reviewer 2 Report (New Reviewer)
Comments and Suggestions for Authors
In this study, the authors explored the influence of different process parameters on the dimensions of the deposited layers, revealing the extent of the impact these parameters have. Additionally, a PSO-XGBoost model was developed to predict the dimensions of the deposited layers, achieving impressive prediction accuracy, which is intriguing. I suggest that the manuscript can be accepted for publication once the authors address the following issues:
(1) When certain specialized terms appear for the first time as acronyms or specific nouns, it is essential to provide the corresponding full English names, such as “MIG”.
(2) The introduction requires further refinement to make the literature review more structured and to emphasize the necessity and novelty of this study.
(3) Some sentences are overly lengthy and complex, which hinder comprehension. Shortening and simplifying these sentences, while maintaining scientific accuracy, will enhance the overall clarity of the manuscript. Such as “Laser arc hybrid welding and laser arc hybrid additive manufacturing exhibit certain commonalities, such as the inverse relationship between melt pool width and depth with respect to travel speed. This observed phenomenon aligns consistently with findings within the laser arc hybrid additive manufacturing.”.
(4) Please check Fig. 3a. The legend and x-axis should be different.
(5) “w” in the legend of Fig. 4c should be “W”.
(6) For Fig. 7 and Fig. 8, the x-axis label may be “Experiment No.” ?
(7) Table 3 and Table 4 have the same title. For better understanding, they should be different.
(8) The conclusions are not concise enough and lack a robust summary of the main findings. A succinct conclusion should be written that highlights the study's primary discoveries and impact, emphasizing its contribution to the field. Addressing these specific issues will significantly enhance the academic quality and readability of the manuscript.
Comments on the Quality of English LanguageSome sentences are too complex and can be improved.
Author Response
Please see the attachment.

Reviewer 3 Report (New Reviewer)
Comments and Suggestions for Authors
The authors used laser-arc hybrid additive manufacturing technology to predict and quantitatively analyze the geometric dimensions of the deposited layer. The influence mechanism of arc current, laser power and scanning speed on the geometric size of the sedimentary layer was analyzed and the relevant theoretical prediction model was constructed. This study has certain theoretical significance. The main problems are described as follows:
(1) How to determine the selection range of laser power and arc current? It should be explained.
(2) The data processing of this paper only analyzes the average value of the height and width of the sedimentary layer from the stable area, which has certain limitations. Laser additive manufacturing technology is easy to produce saddle-shaped morphology characteristics with high at both ends and low in the middle when preparing multi-layer deposited parts. If the author can establish relevant theoretical prediction models for the height changes on both sides and in the middle area, it may have greater research significance.
(3) In this paper, the influence of laser power, scanning speed and arc current on the geometric size of the deposition layer was analyzed from the perspective of molten pool fluidity. So will the pore size changes generated in the process of changing these process parameters also affect the width and height of the sedimentary layer?
Round 2
Reviewer 3 Report (New Reviewer)
Comments and Suggestions for Authors
The authors have made a commendable effort to address the reviewer's concerns within the available space. I recommended to receive this paper.
This manuscript is a resubmission of an earlier submission. The following is a list of the peer review reports and author responses from that submission.
Round 1
Reviewer 1 Report
Comments and Suggestions for Authors
This article discusses the prediction of geometric dimensions of deposited layers in Laser-arc hybrid additive manufacturing (LAHAM). The article begins with an introduction to the background and significance of LAHAM, noting that accurate prediction and control of the geometric dimensions of deposited layers are crucial for maintaining forming accuracy. The research investigates the effects of different process parameters on the width and height of the deposited layers through experiments and in-depth analysis and proposes a prediction model to accurately estimate these dimensions.
Despite some innovation and the model's decent prediction performance, there are several significant issues:
1. In Sections 4.2.1 and 4.2.2, the authors do not clearly state whether the data presented are from the training set or the test set. Moreover, the article does not describe the sampling method of the dataset, which is critical information for this study, and its absence affects the reliability and validity of the results.
2. The presentation of results lacks a clear distinction between the training set and test set, which is crucial for assessing the model's generalizability, and lacks actual experimental and verification processes. This makes it difficult to evaluate the practical application value of the model.
3. Additionally, the information in the figures is incomplete. For example, the response curve shown in Figure 3 does not clearly organize and label the relationship with the X-axis labels.
Comments on the Quality of English LanguageThe English expression has many issues and needs improvement.
Reviewer 2 Report
Comments and Suggestions for Authors
The manuscript entitled "Prediction of geometric dimensions of deposited layer produced using laser-arc hybrid additive manufacturing" demonstrates and discusses an approach to predict the width and height in hybrid arc-laser DED. I recommend rejecting this manuscript for the following reasons:
The first conclusion is trivial and does not contribute to any new knowledge in hybrid arc-laser DED.
The second conclusion is not supported by any proper analysis and evaluation. The predictive performance of three models was compared and the one called PSO-XGBoost was suggested as superior to the other. The suggested model’s hyperparameters were tuned by optimization whereas there is no discussion about any tuning of hyper parameters in the other two, SVR, and Ridge Regression models. Such unfair comparisons are not scientifical and give no valid ground for conclusions.
Comments on the Quality of English LanguageLanguage is OK